# Evidence-based dust exposure prediction and/or control tools in occupational settings: A scoping review protocol

**Gebisa Guyasa Kabito**[1]*, **Yonatal Tefera**[1,2], **Chandnee Ramkissoon**[1], **Sharyn Gaskin**[1]

**1** Adelaide Exposure Science and Health, School of Public Health, The University of Adelaide, Adelaide, South Australia, Australia, **2** Centre for Health in All Policies Research Translation, South Australian Health and Medical Research Institute, Adelaide, South Australia, Australia

* gebisa.kabito@adelaide.edu.au, gebeguyasa4@gmail.com

**Data Availability Statement:** There are no data sets associated with this protocol.

**Funding:** The author(s) received no specific funding for this work.

## Abstract

### Background

Workplace atmospheric exposure monitoring is the standard method to assess and control hazardous dust exposure; however, feasibility and cost constraints often limit its application. In recent decades, evidence-based tools supporting exposure modelling and control banding have been developed to aid in predicting and/or controlling occupational exposure to various contaminants. However, there is limited information on the availability and applicability of evidence-based tools for predicting and/or controlling occupational dust exposure, as well as on the methods for evaluating these tools across different exposure scenarios. Therefore, this planned scoping review aims to identify existing evidence-based tools for dust exposure predicting and/or controlling and to present evaluation approaches.

### Methods

We will employ the scoping review methods developed by the Joanna Briggs Institute (JBI). The search will be conducted on PubMed, Scopus, and Web of Science databases, in addition to grey literature from the National Institute for Occupational Safety and Health (NIOSH) and advanced Google searches. Studies will be included if they report evidence-based tools for predicting and/or controlling dust exposure using quantitative or semi-quantitative designs and provide a detailed explanation of the methods used for tool development. There will be no restrictions on publication date or geographical location; however, only studies published in English will be considered. Studies focusing exclusively on dust exposure in environmental settings will be excluded. Each member of the review team will screen titles, abstracts, and full texts independently and in collaboration, based on the inclusion criteria. The extracted data will encompass details such as author, title, country, accessible platforms, method/tool names, intended users, types of dust, and occupational settings. Descriptions of the identified tools will include numerical data and narrative summaries to ensure a comprehensive overview.

**Competing interests:** The authors have declared that no competing interests exist.

## Trial registration

OSF (https://doi.org/10.17605/OSF.IO/S6EZJ).

## Introduction

The World Health Organization (WHO) defines dust as "solid particles ranging in size from below 1 μm up to at least 100 μm, which may be or become airborne, depending on their origin, physical characteristics, and ambient conditions" [1]. Thus, exposure to dust is typically evaluated based on particle size, which is categorized into coarse ($> 2.5$ μm), fine ($< 2.5$ μm), and ultrafine particles (UFP) ($< 100$ nm) [2]. In occupational settings, exposure to dust primarily arises from mechanical processes such as cutting, breaking, crushing, drilling, abrasive and sand blasting, digging, or hammering [3–5].

Occupational respiratory diseases (ORDs) related to overexposure to dust represent a significant global public health concern, contributing to approximately one-third of all documented work-related mortality [6, 7]. Moreover, the Global Burden of Disease (GBD) reported about 4 million deaths in 2019 [8] and 13.6 million disability-adjusted life years (DALY) due to ORDs [7]. Exposure to dust has also been associated with an increased risk of cardiovascular diseases, with heightened prevalence observed among mine workers and those exposed to silica, diesel exhaust, and inorganic dust, as well as in construction, metal industries, asphalt work, and heavy equipment operation [9, 10].

The prevention of dust-related health burdens critically depends on rigorous exposure assessment and effective dust control measures in the workplace [11]. Exposure assessments, a major component of the occupational health risk assessment process, have been applied across various occupational contexts, leading to the development of numerous definitions and methods (tools) for exposure estimation and control of risks to different toxic agents [12–14]. For example, conventional methods require measurements by trained professionals, such as certified occupational hygienists, who measure dust exposure levels and compare them against established Workplace Exposure Standards (WES) and recommend dust mitigation or control measures accordingly [15, 16]. While the conventional exposure monitoring method is considered the gold standard in dust exposure assessment and control, its application is not always practical for all exposure scenarios due to constraints related to time, expertise, and cost [17–19].

In such scenarios, exposure modelling tools are an alternative method [20]. Considering this, there has been a significant rise in the development and implementation of evidence-based tools in recent decades. In this context, 'evidence-based' refers to tools that are supported by exposure models, exposure data (databases), and control banding techniques for predicting and/or controlling different contaminants in the workplace [21–23]. For example, exposure models are conceptual or mathematical models that enable the estimation of individual exposure parameters based on available input data from specific occupational exposure scenarios [24, 25]. Likewise, control banding employs models that provide control guidance (bands) based on input occupational exposure information [22]. Evidence-based tools can be accessed online through various platforms [26], including web-based interfaces [27, 28], software applications [29], mobile apps [30], and Microsoft Excel [22].

Despite the development of different evidence-based tools for predicting and/or controlling exposure to different contaminants, it is unclear which tools in the literature are specifically relevant for occupational dust exposure. Additionally, there is a lack of information on the evaluation approaches for these tools, which could guide users in selecting the tool for specific dust exposure scenarios.

We conducted a preliminary search across different databases, including PROSPERO, MEDLINE, PUBMED, the Cochrane Database of Systematic Reviews, and JBI Evidence Synthesis, to determine whether the topic has been addressed. We identified one systematic review [31] that synthesized evidence on the reliability of models recommended by the European Chemicals Agency (ECHA), evaluating their precision, accuracy, and robustness in exposure assessment [32]. However, there is no current or ongoing scoping or systematic reviews were found on the evidence-based tools used for occupational dust exposure and the evaluation approaches for these tools. We have chosen a scoping review as a suitable and comprehensive method for synthesizing evidence, as it allows us to examine a diverse body of evidence and outline the fundamental concepts within this research domain [33].

To address this gap, the proposed scoping review will identify existing evidence-based tools for predicting and/or controlling worker dust exposure, as well as explore methodologies/approaches used for evaluating these identified tools. We anticipate that this scoping review will serve as a valuable resource by presenting relevant evidence-based tools for dust exposure prediction and/or control in occupational settings, along with the evaluation approaches for these tools. This will serve as a reference for making informed decisions when selecting the appropriate tools for various dust exposure scenarios.

## Methods

This study will utilize the Joanna Briggs Institute (JBI) method for scoping reviews, recognized for its structured approach to elements such as research questions, inclusion and exclusion criteria, search strategies, and data extraction, among others [34]. The final report of this proposed scoping review will follow the reporting guidance recommended by Preferred Reporting Items for Systematic Reviews and Meta-Analyses Extension for Scoping Reviews (PRISMA-ScR), enhancing transparency through specific reporting criteria and facilitating a thorough and standardized presentation of the scoping review findings (S1 Appendix) [35].

### Protocol and registration

The protocol has been registered on the Open Science Framework (https://doi.org/10.17605/OSF.IO/S6EZJ).

### Review question

The specific research questions for this scoping review are:

1. What evidence-based tools are available in the literature for predicting and/or controlling dust exposure in occupational settings?

2. What evidence is available regarding the evaluation approaches for these tools?

### Eligibility criteria

We utilized the PCC (Population/problem, Concept, and Context) framework as recommended by JBI [34].

### Concept

The core focus of this proposed study is on evidence-based tools for predicting and/or controlling dust exposure, as well as the evaluation approaches relevant to these tools. According to the definition of 'evidence-based' provided in the background section, a tool is deemed evidence-based in this review if it meets specific criteria: it must be incorporated into or based on databases

**Table 1. Summary of problem, concept, and context framework for the evidence-based tools scoping review protocol.**

| Problem | Dust exposure |
|---|---|
| Main concept | • Evidence-based tools for exposure prediction and/or control; and evaluation approaches. |
| Context | • All occupational settings |

(such as silica dust monitoring data from literature, exposure scenarios, measurements, or archived government data) and employ statistical, mathematical, or computational models to predict dust exposure levels and/or recommend control measures (control bands). Additionally, evaluation approaches are operationalized as the quantitative or qualitative methods used to assess the effectiveness of these tools, considering reliability, validity, applicability, and efficacy.

## Problem

The study problem was occupational dust exposure, which includes respirable dust, respirable crystalline silica, and other hazardous dusts such as coal, minerals, wood, metal, and both organic and inorganic dusts generated in the workplace to which workers may be exposed during work activities.

## Context

This proposed scoping review is limited to occupational settings, encompassing a wide range of work environments, including but not limited to construction sites, manufacturing plants, mining, and quarrying. This review will exclude studies conducted solely in environmental settings, which result in exposures that might occur outside of work environments (Table 1).

## Information sources

The proposed scoping review will include peer-reviewed, primary, and grey literature sources that are accessible to the public and that fulfil the eligibility criteria (Table 2).

## Search strategy

The databases that will be searched are PubMed, Scopus, and Web of Science. Grey literature will be sourced through the National Institute for Occupational Safety and Health (NIOSH) platform and advanced Google searches. The search strategy used relevant keywords, subject headings, and MeSH terms specific to each database, incorporating Boolean operators "AND" and "OR" to construct comprehensive search strings. This strategy was developed in

**Table 2. Summary of inclusion and exclusion criteria for the evidence-based tools scoping review protocol.**

| Inclusion criteria | • Publications describing evidence-based tools for dust exposure prediction and/or control will be included. These publications should include a full description of the tool's development process.<br>• Only publications in English language.<br>• All publications without date and location restriction.<br>• Publications must focus on dust exposure in any occupational settings.<br>• Quantitative or semi-quantitative study design. |
|---|---|
| Exclusion criteria | • Publications lacking a full description of the tool's development process.<br>• Publications missing abstract and/or full text.<br>• Publications focusing solely on dust exposure in environmental settings. |

**Table 3. A logic grid of keywords structured by problem, concept, and context framework for the evidence-based tools scoping review protocol.**

| Search guide | #1. Problem | #2. Concept and context |
|---|---|---|
| Phenomena of interest | Dust exposure | Evidence-based tools for prediction and/or control of dust exposure, Evaluation approaches |
| Alternative keywords or Searches in each domain | Dust OR Silica OR Aerosol OR Particulate matter OR wood OR metal OR Coal OR Organic OR inorganic | Web-based OR Online OR Internet OR Device OR Instrument OR technology OR methods OR Models OR risk assessment OR Exposure prediction OR occupational exposure OR occupational exposure assessment OR occupational exposure model OR exposure assessment OR Exposure model OR Exposure model assessment OR exposure measurement OR exposure scenarios OR estimation OR evaluation methods OR Risk management OR exposure control OR risk control OR Exposure control plan OR Control banding OR exposure control banding OR approaches |
| Final search | [1 AND 2] | |

consultation with a qualified librarian from the University of Adelaide. Initially, a preliminary search on PubMed was conducted to identify articles using terms such as dust exposure, exposure models, occupational exposure models, exposure band, exposure scenario, exposure prediction, tools, risk assessment, risk management, control banding, evaluation methods, and exposure control. The PubMed search query was subsequently refined by integrating text words from relevant article titles and abstracts, along with MeSH terms used to classify these articles (S2 Appendix). The 'concept' and 'context' were combined in our search strategy due to a low yield of citations when searched separately. For advanced Google Scholar searches, the same specific terms will be used, and the first 100 articles will be included (S3 Appendix). The search strategy was designed with data charting and literature mapping in mind, as detailed in (Table 3). The search process will be iterative, allowing for the identification and incorporation of additional keywords and relevant search terms as needed.

## Study selection

All records retrieved will be imported into Covidence (www.covidence.org) for screening and removing duplicates. To ensure that all members of the review team (S.G., C.R., Y.T., G.G.K.) are fully familiar with the eligibility criteria and selection process, we will pilot-test 25 articles during the title and abstract screening phase. Following this pilot test, three independent reviewers (G.G.K., C.R, S.G.) will screen all records by titles and abstract against the established eligibility criteria (Table 2). Any discrepancies between the reviewers' assessments will be discussed and resolved, with the criteria refined as necessary. This iterative process will continue until both reviewers reach full agreement on the eligibility of the articles. After the title and abstract screening is completed, full-text screening of the selected publications will proceed. Two reviewers will independently assess the full-text articles to determine their inclusion. The team will then collaborate to reach a consensus on each article. Articles that do not meet the criteria will be excluded, with the reasons for exclusion documented and reported in the final scoping review report.

## Data extraction

We customized the JBI data extraction tool to fit our study objectives, specifying the information required for each category (S4 Appendix). Two independent reviewers will extract data

from the studies included using this tool and will collaborate to finalize the data. During the extraction process, the tool will be modified and revised as necessary to address any issues that arise. All modifications will be documented in the final scoping review report. The extracted data will encompass various study characteristics, including author, title, publication year, and country of origin. Additionally, it will cover details such as accessible platforms, method/tool names, intended users, types of dust, and occupational settings. Furthermore, the data will include information on the description of the methods/tools used, including control bands, databases, exposure bands, input parameters, model types, and evaluation methodologies. If further information is required, the authors of the included studies will be contacted.

## Data analysis and presentation

Extracted data will be analysed using descriptive statistics (frequency and proportion) and a narrative summary. The results will be presented through tables and figures to facilitate clear understanding and contextualization. Additionally, the implications for future research and practical applications will be discussed.

## Ethics and dissemination

Ethical approval is not required since the study is based on publicly available literature. The final report will be disseminated through peer-reviewed journals and relevant conferences.

## Limitations

This planned scoping review will not assess the methodological rigor of the included studies. The diverse terminology used in exposure prediction and dust-related research may result in some relevant studies being overlooked in our search strategy. Finally, only studies published in English will be included in this review.

## Conclusions

This planned scoping review will, to the best of the authors' knowledge, be the first to address existing knowledge gaps in evidence-based tools for dust exposure prediction and/or control in occupational settings. It will systematically identify these tools and their evaluation methods. The findings from this review are expected to serve as a reference for selecting and applying evidence-based dust exposure prediction and/or control tools in various exposure scenarios.

## Deviations from the study protocol

Any deviations from the study protocol will be described in the final report.

## Supporting information

**S1 Appendix. PRISMA-ScR checklist.**
(DOCX)

**S2 Appendix. PubMed search strategy.**
(DOCX)

**S3 Appendix. Search strategy for Google advanced search logic grid.**
(DOCX)

**S4 Appendix. Data extraction tool.**
(DOCX)

## Acknowledgments

The authors would like to thank Vikki Langton, a librarian at The University of Adelaide, for her invaluable assistance in developing a comprehensive search strategy for this proposed scoping review. G.G.K. acknowledges support from the Australian Government Research Training Program (RTP) Scholarship.

## Author Contributions

**Conceptualization:** Yonatal Tefera, Chandnee Ramkissoon, Sharyn Gaskin.

**Investigation:** Gebisa Guyasa Kabito, Yonatal Tefera, Chandnee Ramkissoon, Sharyn Gaskin.

**Methodology:** Gebisa Guyasa Kabito, Yonatal Tefera, Chandnee Ramkissoon, Sharyn Gaskin.

**Supervision:** Yonatal Tefera, Chandnee Ramkissoon, Sharyn Gaskin.

**Writing – original draft:** Gebisa Guyasa Kabito.

**Writing – review & editing:** Gebisa Guyasa Kabito, Yonatal Tefera, Chandnee Ramkissoon, Sharyn Gaskin.

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
