## [Decision Letter · Decision Letter 0]

13 May 2024

PONE-D-24-08686Occupational Dust Exposure Assessment and Control Tools: A Scoping Review ProtocolPLOS ONE

Dear Dr. Kabito,

Thank you for submitting your manuscript to PLOS ONE. After careful consideration, we feel that it has merit but does not fully meet PLOS ONE’s publication criteria as it currently stands. Therefore, we invite you to submit a revised version of the manuscript that addresses the points raised during the review process.

We look forward to receiving your revised manuscript.

Kind regards,

Pisirai Ndarukwa, Ph.D.

Academic Editor

PLOS ONE

Journal Requirements:

Reviewers' comments:

Reviewer's Responses to Questions

**Comments to the Author**

1. Does the manuscript provide a valid rationale for the proposed study, with clearly identified and justified research questions?

Reviewer #1: Partly

2. Is the protocol technically sound and planned in a manner that will lead to a meaningful outcome and allow testing the stated hypotheses?

Reviewer #1: Partly

3. Is the methodology feasible and described in sufficient detail to allow the work to be replicable?

Reviewer #1: No

4. Have the authors described where all data underlying the findings will be made available when the study is complete?

Reviewer #1: Yes

5. Is the manuscript presented in an intelligible fashion and written in standard English?

Reviewer #1: Yes

6. Review Comments to the Author

You may also provide optional suggestions and comments to authors that they might find helpful in planning their study.

**Reviewer #1:** The manuscript under review contains the key components and framework for a productive scoping review mapping existing evidence-based tools for occupational dust exposure assessment and control worthy of consideration for publication.

Overall, there are many strengths to the submission, and the authors should be commended for their proposal of this timely review.

As regards to primary concerns, the following are recommended for consideration:

Across the protocol, I would recommend that greater clarity be given to how the prospective tools are framed: does the tool need to be both an assessment AND a control tool for occupational dust exposure, or can the tool fulfil only one of these functions? If there exists a tool for assessing occupational dust, but it does not contain an element or component for controlling dust, will it still be considered for inclusion (and vice versa)? There are implications of this decision from the title, through the abstract and in particular through the methods section (especially search strategy – currently reads as if you are search for one OR the other, does not have to be both). If a tool must do both assess and control, this will need to be made more explicit in the methods section. Suggest providing an operational definition for what an occupational dust assessment / control tool is

The authors are encouraged to reflect on the purpose and utility of this scoping review once completed – ambiguity appears at various points in the introduction through to the review question. In particular – is this scoping review to map the tools available and the methodologies associated with the literature utilising said tools (as it reads from line 110-114) or is it to examine the evidence to identify evidence that supports the decision-making process for selecting tools (implied in line 83-87, and line 104-106). To simplify, is the review to map the tools that exist or to map the evidence around decision making process for selecting tools or the efficacy of the tools? If it just presenting the tools, as the questions currently read, it is unclear how this will provide the foundation for clear guidance and evidence on the process through which workers etc select the appropriate tools. It can present which tools are being used where, and perhaps what evidence supports their efficacy (if RQ1 is expanded to include this), but it does not extract evidence around the selection process of the tools themselves.

Suggest considering expanding RQ1 to include what the evidence is for either a) the effectiveness of the tools and / or b) the implementation challenges (process evaluation-type variables) associated with using the various tools.

Within methods, suggest parsing out population and context (line 119) into separate paragraphs as you have stated you will use the PCC framework and further develop and clarify the nuances between each of these components, clearly articulating what will be included/excluded under each heading. Table 1 appears incongruent, as it is now ‘problems, concept, context’. Suggest revising to be consistent. With regards Table 1, if dust is extensively searched in concept 1, then adding it to some of the terms in concept 2 (tools) is redundant. The final search at the bottom of table 1 does not appear to be correct, and needs revising – is the final search [1 and 2 and (3a or 3b) and 4]? If so, the two lines above appear superfluous.

It is interesting to read that the authors are proposing to follow both JBI and PRISMA requirements for scoping reviews, and this is slightly confusing to the reviewer. For instance, the framing of the approach under the methods heading from line 115-118, and the methods heading from 135-140 appears redundant and should be clarified – are the approaches complementary or can one be adopted more wholeheartedly?

The development of the search strategy appears fairly adhoc – was there any formal structure used to guide this development? Would there be scope to audit using established tool, such as PRESS guidelines (McGowan et al., 2016) to ensure rigour? Regarding the strategy itself, the fields for the pubmed search strategy have not been specified across each of the lines (in particular 1). There are no MeSH terms specified in 1, have these been considered? Are any date limiters going to be used? Language or other filters? Provide clarity here. There appears to be a discrepancy between Table 1 brainstorm of terms and the search in PubMed, consider aligning. Running this search in PubMed yields a few errors (in particular line 2), but still produces only 166 results. Are you confident that this search will yield sufficient results? This reviewer wonders if the terms around the tools are too specific or maybe overlapping (line 2-3). Search strategy for additional databases and details around searching grey sources would be recommended to include (in particular, how will you search in google scholar).

Through study selection – clarify whether you mean two independent reviewers means that their decisions will be blinded from each other.

It is unclear which types of study designs will be included – assuming it is all quantitative methodologies of tool assessment, but will you also include qualitative evaluations of tools? This should be made explicit within the protocol.

Figure 1 is not needed in a protocol paper and should be removed.

While it is acknowledged that a risk of bias appraisal (quality appraisal) is not always required in scoping reviews, they can be called for in certain instances. In this proposed review, as RQ2 seeks to map the methodologies and approached reported for evaluating occupational dust exposure assessment and control tools, it is recommended that a formal tool be employed to assess the methodological quality of the included studies as this would lend additional rigour and context to the answer for RQ2. If both qual/quant studies are to be included, suggest perhaps the MMAT (Hong 2018)?

Regarding data extraction tool, it appears that what is presented might only apply to RQ1, what data regarding RQ2 will be extracted?

Final minor comments: recommend revising the entire manuscript to ensure the proper tense throughout for a protocol paper, at times ebbing between past (line 86) and future (line 110, for example). Line 139-140 is confusing and requires revision.

7. PLOS authors have the option to publish the peer review history of their article (what does this mean?). If published, this will include your full peer review and any attached files.

Reviewer #1: No

---

## [Author Response · Author response to Decision Letter 0]

14 Aug 2024

Authors’ Response to Reviewers’ Comments 

Manuscript Number: PONE-D-24-08686

Submission Title: Occupational Dust Exposure Assessment and Control Tools: A Scoping Review Protocol

Authors: Gebisa Guyasa Kabito, Yonatal Tefera, Chandnee Ramkissoon, Sharyn Gaskin

16th August 2024

Dear Dr. Pisirai Ndarukwa,

We are thankful for the opportunity to submit a revised version of our manuscript originally titled "Occupational Dust Exposure Assessment and Control Tools: A Scoping Review Protocol" to PLOS ONE (now titled “Evidence-based dust exposure prediction and/or control tools in occupational settings: A scoping review protocol”). We sincerely appreciate the thorough review and the constructive feedback provided by you and the reviewer(s). We have carefully considered all the comments and suggestions and have made comprehensive revisions to the manuscript accordingly. The changes have been tracked for your convenience. We believe that the manuscript has significant improvement as a result of the changes made. 

Our detailed responses to the reviewer’ and editors' comments are provided in the Table of Revisions below, with references corresponding to the track-changed manuscript.

We declare that all authors have read and approved the revised manuscript and it has not been published previously, nor is it being considered by any other Journal. 

Kind regards,

Gebisa Kabito 

Manuscript ID: PONE-D-24-08686 

Table 1: Table of Revisions corresponding to Editor and Reviewer’s comments made to manuscript (line references correspond to the track-changed manuscript)

Editor: Journal requirements Authors’ response Changes to MS 

(using Track Changes)

1. When submitting your revision, we need you to address these additional requirements. Please ensure that your manuscript meets PLOS ONE's style requirements, including those for file naming. The PLOS ONE style templates can be found at 

Thank you for requesting that we ensure our manuscript meets PLOS ONE's style requirements, including those for file naming. We have reviewed the formatting requirements and verified that it conforms to the required styles.

2. When completing the data availability statement of the submission form, you indicated that you will make your data available on acceptance. We strongly recommend all authors decide on a data sharing plan before acceptance, as the process can be lengthy and hold up publication timelines. Please note that, though access restrictions are acceptable now, your entire data will need to be made freely accessible if your manuscript is accepted for publication. This policy applies to all data except where public deposition would breach compliance with the protocol approved by your research ethics board. If you are unable to adhere to our open data policy, please kindly revise your statement to explain your reasoning and we will seek the editor's input on an exemption. Please be assured that, once you have provided your new statement, the assessment of your exemption will not hold up the peer review process. Thank you for pointing out this. We have now amended the data availability statement as requested Data availability

Lines 241

Thank you for the comment. We have addressed this issue accordingly. Supporting information 

Line 340-350

4. While revising your submission, please upload your figure files to the Preflight Analysis and Conversion Engine (PACE) digital diagnostic tool, https://pacev2.apexcovantage.com/. PACE helps ensure that figures meet PLOS requirements. To use PACE, you must first register as a user. Registration is free. Then, login and navigate to the UPLOAD tab, where you will find detailed instructions on how to use the tool. If you encounter any issues or have any questions when using PACE, please email PLOS at figures@plos.org. Please note that Supporting Information files do not need this step. Not applicable 

Reviewer #1 

The manuscript under review contains the key components and framework for a productive scoping review mapping existing evidence-based tools for occupational dust exposure assessment and control worthy of consideration for publication. Overall, there are many strengths to the submission, and the authors should be commended for their proposal of this timely review.

As regards to primary concerns, the following are recommended for consideration:

1. Across the protocol, I would recommend that greater clarity be given to how the prospective tools are framed: does the tool need to be both an assessment AND a control tool for occupational dust exposure, or can the tool fulfil only one of these functions? Authors agree. Accordingly, we have clarified the focus of the study. In our manuscript, the evidence-based tools refer to both functions: as exposure prediction components, exposure control components, or both. We have operationalized and applied necessary changes throughout the manuscript to ensure clarity, as per the reviewer's recommendations. Abstract section

(Line 36- 37)

Introduction section

(Line 79-81)

Method section 

(Line 132-137) 

2. If there exists a tool for assessing occupational dust, but it does not contain an element or component for controlling dust, will it still be considered for inclusion (and vice versa)? Thank you for your question. Yes, tools for assessing occupational dust exposure will be considered for inclusion in our study even if they do not include components specifically for controlling dust. Similarly, tools focused solely on dust control will also be eligible for consideration. Our aim is to comprehensively review all relevant tools related to occupational dust exposure, whether they address assessment, control, or both aspects. We have added information in the text to clarify this point . method section 

(Line 132-137) 

3. There are implications of this decision from the title, through the abstract and in particular through the methods section (especially search strategy – currently reads as if you are search for one OR the other, does not have to be both). If a tool must do both assess and control, this will need to be made more explicit in the methods section. Authors agree. 

We acknowledge that our current approach implies searching for tools that fulfill either assessment or control functions. Consequently, we have made necessary modifications throughout, including the title, inclusion criteria and search strategy to require tools that perform either assessment, control, or both. Table 2

Line 153

Table 3

Line 177

Appendix II

Line 378

4. Suggest providing an operational definition for what an occupational dust assessment / control tool is Authors agree, and we have now provided an operational definition as suggested. method section 

(Line 132-137) 

5. The authors are encouraged to reflect on the purpose and utility of this scoping review once completed – ambiguity appears at various points in the introduction through to the review question. In particular – is this scoping review to map the tools available and the methodologies associated with the literature utilising said tools (as it reads from line 110-114) or is it to examine the evidence to identify evidence that supports the decision-making process for selecting tools (implied in line 83-87, and line 104-106). To simplify, is the review to map the tools that exist or to map the evidence around decision making process for selecting tools or the efficacy of the tools? Authors accept the suggestion. 

Accordingly, we have now modified and provided a clear focus of the scoping review throughout the manuscript, which is to map existing tools available for occupational dust prediction and/ or control, and to present evaluation approaches applied to these identified tools. 

Introduction 

Section 

103-105

Method section

Line 123-125

6. If it just presenting the tools, as the questions currently read, it is unclear how this will provide the foundation for clear guidance and evidence on the process through which workers etc select the appropriate tools. It can present which tools are being used where, and perhaps what evidence supports their efficacy (if RQ1 is expanded to include this), but it does not extract evidence around the selection process of the tools themselves. Authors partly agree. 

Authors acknowledge the importance of extracting direct evidence around the selection process of the tools, but this aspect is not included in our review, and will be addressed in future work.

However, this scoping review is designed to do more than just list available tools. It incorporates evaluation approaches that detail the validity, applicability, and suitability of each tool for specific types of dust in various occupational settings. 

By systematically mapping both the tools and their evaluation approaches, we plan to provide a detailed analysis that supports or provides a basis for further work related to informed decision-making. 

7. Suggest considering expanding RQ1 to include what the evidence is for either a) the effectiveness of the tools and / or b) the implementation challenges (process evaluation-type variables) associated with using the various tools. We thank the reviewer for their valuable suggestion. The authors believe this is a very important point and will aim to address it in a separate scope of work but will not be expanding the current scope.

We anticipate being able to address certain aspects of this point through our examination of existing evaluation approaches, which will include validation methods applied to these identified tools. 

Method section

Line 137-139

8. Within methods, suggest parsing out population and context (line 119) into separate paragraphs as you have stated you will use the PCC framework and further develop and clarify the nuances between each of these components, clearly articulating what will be included/excluded under each heading. That you for your suggestion. We agree.

Accordingly, we made necessary amendments to clarify the nuances of the framework. Table 2

Line 153

9. Table 1 appears incongruent, as it is now ‘problems, concept, context’. Suggest revising to be consistent. With regards Table 1, if dust is extensively searched in concept 1, then adding it to some of the terms in concept 2 (tools) is redundant. We agree. 

We have revised Table 1 to avoid redundancy in the search terms and to ensure consistency with the structure of 'problems, concept, context'. 

Method section

Table 3

Line 177

10. The final search at the bottom of table 1 does not appear to be correct, and needs revising – is the final search [1 and 2 and (3a or 3b) and 4]? If so, the two lines above appear superfluous. Authors agree. 

Accordingly, we have now revised as per suggestion. Method section

Table 3

Line 177

11. It is interesting to read that the authors are proposing to follow both JBI and PRISMA requirements for scoping reviews, and this is slightly confusing to the reviewer. For instance, the framing of the approach under the methods heading from line 115-118, and the methods heading from 135-140 appears redundant and should be clarified – are the approaches complementary or can one be adopted more wholeheartedly? Authors agree. 

Accordingly, we have now amended and provided clarity. The JBI framework provides a structured approach tailored for scoping reviews, emphasizing clarity in research questions, comprehensive search strategies, and systematic data extraction. To compliment, PRISMA-ScR enhances transparency by delineating specific reporting criteria, facilitating a thorough and standardized presentation of scoping review findings. Method section

Line 111-117

12. The development of the search strategy appears fairly adhoc – was there any formal structure used to guide this development? Thank you for your question. 

Yes, we followed the JBI framework. We have clarified this point in the manuscript.

 Line 111-117

13. Would there be scope to audit using established tool, such as PRESS guidelines (McGowan et al., 2016) to ensure rigour? Thank you for your question.

Yes, we can audit our review using the PRESS checklist, and have added this element to the manuscript. Appendix I

Line 362

14. Regarding the strategy itself, the fields for the pubmed search strategy have not been specified across each of the lines (in particular 1). There are no MeSH terms specified in 1, have these been considered? Are any date limiters going to be used? Language or other filters? Provide clarity here. Authors agree.

We have now made the necessary amendments to help clarify the search strategy. Appendix II

Line 378

15. There appears to be a discrepancy between Table 1 brainstorm of terms and the search in PubMed, consider aligning. Authors agree. 

Accordingly, we have now revised for better alignment of Table 1 (now Table 3) brainstorm of terms and the search in PubMed. Table 3

Line 177

Appendix II 

Line 378

16. Running this search in PubMed yields a few errors (in particular line 2), but still produces only 166 results. Are you confident that this search will yield sufficient results? Authors agree. 

We have corrected the error and amended the search strategy as suggested. Now, we are confident that with these refinements, the search yields 890 retrieved citations, providing sufficient citations for the scoping review from a single database. Appendix I 

Line 521

17. This reviewer wonders if the terms around the tools are too specific or maybe overlapping (line 2-3). Thank you for the question.

We have now provided clarity to address any potential overlap or specificity issues related to terms throughout the search strategy Table 3

Line 177

18. Search strategy for additional databases and details around searching grey sources would be recommended to include (in particular, how will you search in google scholar). Authors agree.

We have now provided the search strategy for grey literature e.g., the NIOSH database and advanced Google Scholar Appendix III

Line 387

19. Through study selection – clarify whether you mean two independent reviewers means that their decisions will be blinded from each other. 

Authors agree. We have made the necessary clarity edits, as outlined below.

Accordingly, in our study selection process, two independent reviewers will assess articles separately to ensure unbiased evaluation. While they will review articles independently, their decisions will not be blinded from each other; instead, they will collaborate to reach a consensus on which articles meet the inclusion criteria. Study selection

Line 190-191

20. It is unclear which types of study designs will be included – assuming it is all quantitative methodologies of tool assessment, but will you also include qualitative evaluations of tools? This should be made explicit within the protocol. Authors agree.

Accordingly, we have now modified the protocol to include eligibility criteria specifying which study designs will be considered Table 2

Line 153

21. Figure 1 is not needed in a protocol paper and should be removed. Authors agree. We have removed Figure 1 as suggested. 

22. While it is acknowledged that a risk of bias appraisal (quality appraisal) is not always required in scoping reviews, they can be called for in certain instances. In this proposed review, as RQ2 seeks to map the methodologies and approached reported for evaluating occupational dust exposure assessment and control tools, it is recommended that a formal tool be employed to assess the methodological quality of the included studies as this would lend additional rigour and context to the answer for RQ2. If both qual/quant studies are to be included, suggest perhaps the MMAT (Hong 2018)? We thank the reviewer for their valuable suggestion

---

## [Decision Letter · Decision Letter 1]

22 Aug 2024

Evidence-based dust exposure prediction and/or control tools in occupational settings: A scoping review protocol

PONE-D-24-08686R1

Dear Dr. Kabito,

We’re pleased to inform you that your manuscript has been judged scientifically suitable for publication and will be formally accepted for publication once it meets all outstanding technical requirements.

Kind regards,

Pisirai Ndarukwa, Ph.D.

Academic Editor

PLOS ONE

Additional Editor Comments (optional):

Reviewers' comments:

Reviewer's Responses to Questions

**Comments to the Author**

1. Does the manuscript provide a valid rationale for the proposed study, with clearly identified and justified research questions?

Reviewer #1: Yes

2. Is the protocol technically sound and planned in a manner that will lead to a meaningful outcome and allow testing the stated hypotheses?

Reviewer #1: Yes

3. Is the methodology feasible and described in sufficient detail to allow the work to be replicable?

Reviewer #1: Yes

4. Have the authors described where all data underlying the findings will be made available when the study is complete?

Reviewer #1: Yes

5. Is the manuscript presented in an intelligible fashion and written in standard English?

Reviewer #1: Yes

6. Review Comments to the Author

You may also provide optional suggestions and comments to authors that they might find helpful in planning their study.

Reviewer #1: The authors are commended for their thoughtful consideration of feedback provided, and wished every success with the completion of the review.

7. PLOS authors have the option to publish the peer review history of their article (what does this mean?). If published, this will include your full peer review and any attached files.

Reviewer #1: No

---

## [Editor Report · Acceptance letter]

28 Aug 2024

PONE-D-24-08686R1 

PLOS ONE

Dear Dr. Kabito, 

I'm pleased to inform you that your manuscript has been deemed suitable for publication in PLOS ONE. Congratulations! Your manuscript is now being handed over to our production team.

Kind regards, 

on behalf of

Prof Pisirai Ndarukwa 

Academic Editor

PLOS ONE